# ONLINE SEMI-SUPERVISED LEARNING WITH BANDIT FEEDBACK

**Mikhail Yurochkin, Sohini Upadhyay, Djallel Bouneffouf, Mayank Agarwal, Yasaman Khazaeni**
IBM Research AI
Cambridge, MA, USA and Yorktown Heights, NY, USA
`{firstname.lastname}@ibm.com`

## ABSTRACT

We formulate a new problem at the intersection of semi-supervised learning and contextual bandits, motivated by several applications including clinical trials and dialog systems. We demonstrate how contextual bandit and graph convolutional networks can be adjusted to the new problem formulation. We then take the best of both approaches to develop multi-GCN embedded contextual bandit. Our algorithms are verified on several real world datasets.

## 1 INTRODUCTION

We formulate the problem of Online Partially Rewarded (OPR) learning. Our problem is a synthesis of the challenges often considered in the semi-supervised and contextual bandit literature. Despite a broad range of practical cases, we are not aware of any prior work addressing each of the corresponding components.

*Online*: data incrementally collected and systems are required to take an action before they are allowed to observe any feedback from the environment.

*Partially*: oftentimes there is no environment feedback available, e.g. a missing label

*Rewarded*: instead of the true label, we can only hope to observe feedback indicating whether our prediction is good or bad (1 or 0 reward), the latter case obscuring the true label for learning.

Practical scenarios that fall under the umbrella of OPR range from clinical trials to dialog orchestration. In clinical trials, reward is partial, as patients may not return for followup evaluation. When patients do return, if feedback on their treatment is negative, the best treatment, or true label, remains unknown. In dialog systems, a user's query is often directed to a number of domain specific agents and the best response is returned. If the user provides negative feedback to the returned response, the best available response is uncertain and moreover, users can also choose to not provide feedback.

In many applications, obtaining labeled data requires a human expert or expensive experimentation, while unlabeled data may be cheaply collected in abundance. Learning from unlabeled observations is the key challenge of semi-supervised learning (Chapelle et al., 2009). We note that the problem of online semi-supervised leaning is rarely considered, with few exceptions (Yver, 2009; Valko et al., 2012). In our setting, the problem is further complicated by the bandit-like feedback in place of labels, rendering existing semi-supervised approaches inapplicable. We will however demonstrate how one of the recent approaches, Graph Convolutional Networks (GCN) (Kipf & Welling, 2016), can be extended to our setting.

The multi-armed bandit problem provides a solution to the exploration versus exploitation trade-off while maximizing cumulative reward in an online learning setting. In Linear Upper Confidence Bound (LINUCB) (Li et al., 2010; Chu et al., 2011) and in Contextual Thompson Sampling (CTS) (Agrawal & Goyal, 2013), the authors assume a linear dependency between the expected reward of an action and its context. However, these algorithms assume that the bandit can observe the reward at each iteration. Several authors have considered variations of partial/corrupted rewards (Bartók et al., 2014; Gajane et al., 2016), but the case of entirely missing rewards has not been studied to the best of our knowledge.

The rest of the paper is structured as follows. In section 2, we formally define the Online Partially Rewarded learning setup and present two extensions to GCN to suit our problem setup. Section 3

presents quantitative evidence of these methods applied to four datasets and analyses the learned latent space of these methods.

## 2 METHODS

We first formally define each of the OPR keywords:

*Online*: at each step $t = 1, \ldots, T$ we observe observation $x_t$ and seek to predict its label $\hat{y}_t$ using $x_t$ and possibly any information we had obtained prior to step $t$.

*Partially*: after we make a prediction $\hat{y}_t$, the environment may not provide feedback (we will use -1 to encode its absence) and we must proceed to step $t + 1$ without knowledge of the true $y_t$.

*Rewarded*: suppose there are $K$ possible labels $y_t \in \{1, \ldots, K\}$. The environment at step $t$ will not provide true $y_t$, but instead a response $h_t \in \{-1, 0, 1\}$, where $h_t = 0$ indicates $\hat{y}_t \neq y_t$ and $h_t = 1$ indicates $\hat{y}_t = y_t$ (-1 indicates missing response).

### 2.1 REWARDED ONLINE GCN

Rewarded Online GCN (ROGCN) is a natural extension of GCN, adapted to the online, partially rewarded setting along with a potential absence of true graph information. We assume availability of a small portion of data and labels (size $T_0$) available at the start, $X_0 \in \mathbb{R}^{T_0 \times D}$ and $y_0 \in \{-1, 1, \ldots, K\}^{T_0}$. When there is no graph available we can construct a $k$-NN graph ($k$ is a parameter chosen a priori) based on similarities between observations - this approach is common in convolutional neural networks on feature graphs (Henaff et al., 2015; Defferrard et al., 2016) and we adopt it here for graph construction between observations $X_0$ to obtain graph adjacency $A_0$. Using $X_0, y_0, A_0$, we can train GCN with $L$ hidden units (a parameter chosen a priori) to obtain initial estimates of hidden layer weights $W_1 \in \mathbb{R}^{D \times L}$ and softmax weights $W_2 \in \mathbb{R}^{L \times K}$. Next we start to observe the stream of data — as new observation $x_t$ arrives, we add it to the graph and data matrix, and append -1 (missing label) to $y$. Then we run additional training steps of GCN and output a prediction to obtain environment response $h_t \in \{-1, 0, 1\}$. Here 1 indicates correct prediction, hence we

---

**Algorithm 1** GCNUCB

1: **Input:** $W_1^{(k)}, W_2^{(k)}, \mathcal{C}_k, r_{\cdot,k}, y_0^{(k)} \ \forall k, \ X_0, \hat{A}_0, \alpha$
2: Set $y^{(k)} = y_0^{(k)} \ \forall k, \ X = X_0, \hat{A} = \hat{A}_0$
3: **for** $t = T_0 + 1$ **to** $T$ **do**
4:     Append $x_t$ to $X$, -1 to each of $y^{(1)}, \ldots, y^{(K)}$
5:     Update $\hat{A}$ with new edges using $x_t$
6:     Update GCN weights $W_1^{(k)}, W_2^{(k)}$ using $X, \hat{A}, y^{(k)}, \forall k$
7:     Retrieve GCN embeddings $g(X)^{(k)}$
8:     $\mathbf{A}_k = \sum_{t \in \mathcal{C}_k} g(X)_t^{(k)} g(X)_t^{(k)^\top}$
9:     $\theta_k = \mathbf{A}_k^{-1} \sum_{t \in \mathcal{C}_k} r_{t,k} g(X)_t^{(k)}$
10:     $\theta_k = \theta_k / \|\theta_k\|_2$
11:     $\mu_k = \theta_k^\top g(X)_t^{(k)}$
12:     $\sigma_k = \alpha \sqrt{g(X)_t^{(k)^\top} \mathbf{A}_k^{-1} g(X)_t^{(k)}}$
13:     Predict $\hat{y}_t = \operatorname{argmax}_k(\mu_k + \sigma_k)$ and observe $h_t$
14:     **if** $h_t = 1$ **then**
15:         $\forall k, \ y_t^{(k)} = 1$ if $\hat{y}_t = k$ , 0 otherwise
16:         Append $t$ to each $\mathcal{C}_k$ and 1 to $r_{\cdot,k}$ if $\hat{y}_t = k$ and 0 otherwise
17:     **else if** $h_t = 0$ (learning from mistakes) **then**
18:         $y_t^{(\hat{y}_t)} = 0$. Append $t$ to $\mathcal{C}_{\hat{y}_t}$ and 0 to $r_{\cdot,\hat{y}_t}$
19:     **else if** $h_t = -1$ (imputing) **then**
20:         Append $t$ to $\mathcal{C}_{\hat{y}_t}$, output of $\hat{y}_t$-th GCN to $r_{\cdot,\hat{y}_t}$

---

include it to the set of available labels for future predictions; 0 indicates wrong prediction and -1 an absence of a response, in the later two cases we continue to treat the label of $x_t$ as missing.

### 2.2 MULTI-GCN EMBEDDED UCB

ROGCN is unable to learn from missclassified observations and has to treat them as missing labels. The bandit perspective allows one to learn from missclassfied observations, i.e. when the environment response $h_t = 0$, and the neural network perspective facilitates learning better features such that linear classifier is sufficient. This observation motivates us to develop a more sophisticated synthesis of GCN and LINUCB approaches, where we can combine advantages of both perspectives. Notice that if $K = 2$, a $h_t = 0$ environment response identifies the correct class, hence the OPR reduces to online semi-supervised learning for which GCN can be trivially adjusted using ideas from ROGCN. To take advantage of this for $K > 2$, we propose to use a suite of class specific GCNs,

where the hidden layer representation from the $k$-th class GCN, i.e. $g(X)^{(k)} = \hat{A}\,\text{ReLU}(\hat{A}XW_1^{(k)})$ and $g(X)_t^{(k)}$ denotes the embedding of observation $x_t$, is used as context by the contextutal bandit for the predictions of the $k$-th arm. Based on the environment response to the prediction, we update the labels and the reward information to reflect a correct, incorrect, or a missing environment response. The reward is imputed from the corresponding GCN when the response is missing.

As we add new observation $x_{t+1}$ to the graph and update weights of the GCNs, the embedding of the previous observations $x_1, \ldots, x_t$ evolves. Therefore instead of dynamically updating bandit parameters, we maintain a set of indices for each of the arms $\mathcal{C}_k = \{t : \hat{y}_t = k \text{ or } h_t = 1\}$ and use observations and responses from only these indices to update the corresponding bandit parameters.

Similar to ROGCN, we can use a small amount of data $X_0$ and labels $y_0$ converted to binary labels $y_0^{(k)} \in \{-1, 0, 1\}^{T_0}$ (as before -1 encodes missing label) for each class $k$ to initialize GCNs weights $W_1^{(k)}, W_2^{(k)}$ for $k = 1, \ldots, K$. We present the GCNUCB in Algorithm 1, where $r_{t,k} \in [0, 1]$ denotes the reward observed or imputed at step $t$ for arm $k$ as described in the algorithm.

## 3 EXPERIMENTS

In this section we compare baseline method LINUCB which ignores the data with missing rewards to ROGCN and GCNUCB. We consider four different datasets: CNAE-9 and Internet Advertisements from the the UCI Machine Learning Repository[1], Cora [2], and Warfarin (Sharabiani et al., 2015). Cora is naturally a graph structured data which can be utilized by ROGCN and GCNUCB. For other datasets we use a 5-NN graph built online from the available data as follows.

Suppose at step $t$ we have observed data points $x_i \in \mathbb{R}^D$ for $i = 1, \ldots, t$. Weights of the similarity graph computed as follows: $A_{ij} = \exp\left(\frac{\|x_i - x_j\|_2^2}{\sigma^2}\right)$. As it was done by Defferrard et al. (2016) we set $\sigma = \frac{1}{t}\sum_{i=1}^{t} d(i, i_k)$, where $d(i, i_k)$ denotes $L_2$ distance between observation $i$ and its $k$-th nearest neighbour indexed $i_k$. The k-NN adjacency $A$ is obtained by setting all but $k$ (excluding itself) corresponding closest entries of $A_{ij}$, $i, j = 1, \ldots, t$ to 0 and symmetrizing. Then, as in Kipf & Welling (2016), we add self connections and row normalize $\hat{A} = (\mathcal{D} + \mathcal{I}_T)^{-1/2}(A + \mathcal{I}_T)(\mathcal{D} + \mathcal{I}_T)^{-1/2}$, where $\mathcal{D}_{ii} = \sum_{j=1}^{T} A_{ij}$ is the diagonal matrix of node degrees.

For pre-processing we discarded features with large magnitudes (3 features in Internet Advertisements and 2 features in Warfarin) and row normalized all observations to have unit $l_1$ norm. For all the methods that use GCN, we use 16 hidden units for GCN, and use Adam optimizer with a learning rate of 0.01, and regularization strength of 5e-4, along with a dropout of 0.5.

To simulate the OPR setting, we randomly permute the order of the observations in a dataset and remove labels for 25% and 75% of the observations chosen at random. For all methods we consider initial data $X_0$ and $y_0$ to represent a single observation per class chosen randomly ($T_0 = K$). At a step $t = T_0 + 1, \ldots, T$ each algorithm is given a feature vector $x_t$ and is ought to make a prediction $\hat{y}_t$. The environment response $h_t \in \{-1, 0, 1\}$ is then observed and algorithms moves onto step $t + 1$. To compare performance of different algorithms at each step $t$ we compare $\hat{y}_t$ to true label $y_t$ available from the dataset (but concealed from the algorithms themselves) to evaluate running accuracy.

Table 1: Total average accuracy

| 25% Missing labels | | | | |
|---|---|---|---|---|
| | CNAE-9 | Internet Ads | Warfarin | Cora |
| LINUCB | $67.57 \pm 2.90$ | $90.08 \pm 0.64$ | $53.70 \pm 0.77$ | $38.06 \pm 3.45$ |
| ROGCN | $64.73 \pm 2.67$ | $88.22 \pm 1.73$ | $47.72 \pm 9.40$ | $48.57 \pm 7.75$ |
| GCNUCB | $\mathbf{77.10 \pm 1.89}$ | $\mathbf{93.14 \pm 0.39}$ | $\mathbf{55.19 \pm 3.40}$ | $\mathbf{66.01 \pm 1.35}$ |
| 75% Missing labels | | | | |
| | CNAE-9 | Internet Ads | Warfarin | Cora |
| LINUCB | $61.67 \pm 3.16$ | $86.66 \pm 0.99$ | $52.99 \pm 2.61$ | $33.92 \pm 0.04$ |
| ROGCN | $65.67 \pm 5.28$ | $88.31 \pm 1.81$ | $47.48 \pm 5.41$ | $49.63 \pm 5.06$ |
| GCNUCB | $\mathbf{70.82 \pm 2.33}$ | $\mathbf{91.45 \pm 0.89}$ | $\mathbf{53.31 \pm 2.98}$ | $\mathbf{58.29 \pm 2.80}$ |

For GCNUCB we use baseline LINUCB for first 300 steps, and for both we use exploration-exploitation trade-off parameter $\alpha = 0.25$. Results are summarized in Table 1. Since ordering of the data can affect the problem difficulty, we performed 10 data resampling for each setting to obtain error margins. GCNUCB outperforms the LINUCB baseline and ROGCN in all of the ex-

[1]https://archive.ics.uci.edu/ml/datasets.html
[2]https://people.cs.umass.edu/ mccallum/data.html

periments, validating our intuition that a method synthesizing the exploration capabilities of bandits coupled with the effective feature representation power of neural networks is the best solution to the OPR problem. We see the greatest increase in accuracy between GCNUCB and the alternative approaches on the Cora dataset which has a natural adjacency matrix. This suggests that GCNUCB has a particular edge in OPR applications with graph structure. Such problems are ubiquitous. Consider our motivating example of dialog systems - for dialog systems deployed in social network or workplace environments, there exists graph structure between users, and user information can be considered alongside queries for personalization of responses.

**Visualizing GCNUCB context space.** Recall that the context for each arm of GC-NUCB is provided by the corresponding binary GCN hidden layer. The motivation for using binary GCNs to provide the context to LINUCB is the ability of GCN to construct more powerful features using graph convolution and neural networks expressiveness.

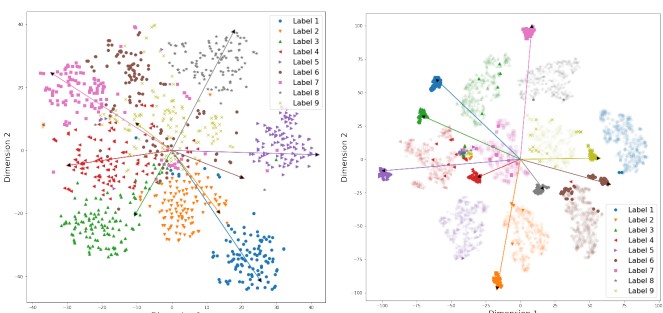

To see how this procedure improves upon the baseline LINUCB utilizing input features as context, we project the context and the corresponding bandit weight vectors, $\theta_1, \ldots, \theta_K$, for both LINUCB and GCNUCB to a 2-dimensional space using t-SNE (Maaten & Hinton, 2008). In this experiment we analyzed CNAE-9 dataset with 25% missing labels. Recall that the bandit makes prediction based on the upper confidence bound of the regret: $\operatorname{argmax}_k(\theta_k^\top x_{k,t} + \sigma_k)$ and that $x_{k,t} = x_t \; \forall k =$

Figure 1: t-SNE embeddings of context and bandit weight vectors for LINUCB

Figure 2: t-SNE embeddings of context and bandit weight vectors for GCNUCB

$1, \ldots, K$ for LINUCB and $x_{k,t} = g(X)_t^{(k)}$ for GCNUCB. To better visualize the quality of the learned weight vectors, for this experiment we set $\alpha = 0$ and hence $\sigma_k = 0$ resulting in a greedy bandit, always selecting an arm maximizing expected reward $\theta_k^\top x_{t,k}$. In this case, a good combination of contexts and weight vectors is the one where observations belonging to the same class are well clustered and corresponding bandit weight vector is directed at this cluster.

For LINUCB (Figure 1, 68% accuracy) the bandit weight vectors mostly point in the direction of their respective context clusters, however the clusters themselves are scattered, thereby inhibiting the capability of LINUCB to effectively distinguish between different arms given the context. In the case of GCNUCB (Figure 2, 77% accuracy) the context learned by each GCN is tightly clustered into two distinguished regions - one with context for corresponding label and binary GCN when it is the correct label (points with bolded colors), and the other region with context for the label and GCN when a different label is correct (points with faded colors). The tighter clustered contexts allow GCNUCB to effectively distinguish between different arms by assigning higher expected reward to contexts from the correct binary GCN than others, thereby resulting in better performance of GCNUCB than other methods.

## 4 CONCLUSION AND DISCUSSION

We have defined and studied the problem of Online Partially Rewarded (OPR) learning, which combines challenges from semi-supervised learning and multi-armed contextual bandits. Our main contribution, GCNUCB algorithm, is the efficient synthesis of the strengths of the two approaches. Our experiments show that GCNUCB, which combines feature extraction capability of the graph convolution neural networks and natural ability of contextual bandits to handle online learning with reward (instead of labels), is the best approach for OPR across a LINUCB baseline and our proposed GCN extension, ROGCN. In our current implementation of GCNUCB we use all of the data seen so far to update parameters. This may not be a permissible choice for large data sizes, however some of the recent work (Hamilton et al., 2017; Chen et al., 2018) has already proposed variants of the mini-batch GCN training, enabling us to make GCNUCB applicable to larger datasets in future work. Theoretical studies of our algorithms will also follow in future work.

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
