# OpenReview forum: "Online Semi-Supervised Learning with Bandit Feedback"
_ICLR.cc/2019/Workshop/LLD — LLD 2019_

### Official Review · AnonReviewer2 · 2019-04-07
**Interesting setting but insufficient positioning in the literature**

**Rating:** 2
**Confidence:** 2

**Review:**

This paper introduces the "Online Partially Rewarded" setting, which appears to be interesting both from a practical and a theoretical point of view. They then propose two approaches based on GCN, which maintain a similarity graph between observations to help predictions. As Rewarded Online GCN only cannot deal with missclassified observations, the authors propose to combine a multi-armed bandit approach with multiple GCN embedding.

The resulting algorithm maintains k different embeddings for the whole dataset, which are recomputed at each iteration (new observation t). This seems to be costly: would it be possible to update only a few last observations? As noted by the authors, this prevents them to apply their algorithm to bigger datasets. A discussion on the performance would still be a plus.

The detail of the algorithm shows that the proposed algorithm does not account for delayed environment response. This seems to be an important limitation, not implied by the OPR setting: at iteration t, the algorithm has to predict y_t and then observes response h_t. If h_t is missing, the corresponding context is set forever, and a late h_t will just be ignored. Recent works in the bandit community propose frameworks which seem to be more general (see e.g. Bandits with Delayed Anonymous Feedback by Pike-Burke et al. or Online Learning under Delayed Feedback by Joulani et al.).

Experiments are conducted on real datasets but with a simulated OPR setting, which is a slight limitation. However, only the two proposed methods and a standard algorithm (LINUCB) of the literature are compared. The results are very encouraging, especially, as noted by the authors, when a natural graph structure is present in the data; but the LINUCB baseline is not adapted to the OPR setting and thus the experiments do not provide enough evidence to back the proposed approach. Here again, I would suggest comparisons with more recent works, such as Variational Thompson Sampling for Relational Recurrent Bandits by Lamprier et al.

---

### Official Review · AnonReviewer1 · 2019-04-10
**Review for paper: Online Semi-Supervised Learning with Bandit Feedback**

**Rating:** 3
**Confidence:** 2

**Review:**

The paper considers a multi-armed bandit problem in which on some iterations, the reward from the environment might be
completely missing. To the rescue, a modified LinUCB-type algorithm is then proposed. Experimental results seam sound.
The authors show that their proposal outpowers a blind LinUCB algorithm. The t-SNE plots (Figs 1 and 2) also show that
the embeddings learned by the proposed model are more compactly clustered w.r.t the true labels.

The paper is well-written and the movitated problems are well described.

The paper considers a multi-armed bandit problem in which on some iterations, the reward from the environment might be
completely missing. To the rescue, a modified LinUCB-type algorithm is then proposed. Experimental results seam sound.
The authors show that their proposal outpowers a blind LinUCB algorithm. The t-SNE plots (Figs 1 and 2) also show that
the embeddings learned by the proposed model are more compactly clustered w.r.t the true labels.

The paper is well-written and the movitated problems are well described.

---

### Decision · Program_Chairs · 2019-04-16
**Acceptance Decision**

Accept